Effect of sodium formate and lactic acid bacteria treated rye silage on methane yield and energy balance in Hanwoo steers

Choi Yongjun 1
Kim Jayeon 2
Bang Geumhwi 3
Kim Nayeon 4
Thirugnanasambantham Krishnaraj 5 6
Lee Sangrak 7
Kim Kyoung Hoon 8 9
Bharanidharan Rajaraman 9 bharanidharan7@snu.ac.kr
1 School of Animal Life Convergence Science, Hankyung National University , Anseong, Gyeonggi-do , South Korea
2 Cargill Agri Purina Inc. , Pyeongtaek, Gyeonggi-do , South Korea
3 Farmsco Co., Ltd. , Anseong, Gyeonggi-do , South Korea
4 Asia Pacific Ruminant Institute , Icheon, Gyeonggi-do , South Korea
5 Pondicherry Centre for Biological Science and Educational Trust , Puducherry , India
6 Department of Biotechnology, Saveetha Institute of Medical and Technical Sciences , Chennai , India
7 Department of Animal Science and Technology, Konkuk University , Seoul , South Korea
8 Department of International Agricultural Technology, Graduate School of International Agricultural Technology, Seoul National University , Pyeongchang, Gwangwon-do , South Korea
9 Department of Eco-friendly Livestock Science, Institutes of Green Bio Science and Technology, Seoul National University , Pyeongchang, Gwangwon-do , South Korea
Gillespie Joseph
Electronic publication date: 2024 Sep 5
Publication date: 2024
Volume: 12
Electronic Location ID: e17920
Received 2023 Dec 13; Accepted 2024 Jul 23
Copyright: © 2024 Choi et al.
Copyright year: 2024
Copyright holder: Choi et al.
License: This is an open access article distributed under the terms of the Creative Commons Attribution License, which permits unrestricted use, distribution, reproduction and adaptation in any medium and for any purpose provided that it is properly attributed. For attribution, the original author(s), title, publication source (PeerJ) and either DOI or URL of the article must be cited.
License URL: https://creativecommons.org/licenses/by/4.0/

Keywords: Lactic acid inoculant, Sodium formate, Acid-based additives, Silage, Methane production, Hanwoo.

Funding: Korea Institute of Planning and Evaluation for Technology in Food, Agriculture and Forestry (IPET) Ministry of Agriculture, Food and Rural Affairs (MAFRA) RS-2024-00400922 This work was supported by Korea Institute of Planning and Evaluation for Technology in Food, Agriculture and Forestry (IPET) through Agriculture and Food Convergence Technologies Program for Research Manpower development, funded by Ministry of Agriculture, Food and Rural Affairs (MAFRA)(RS-2024-00400922). The funders had no role in study design, data collection and analysis, decision to publish, or preparation of the manuscript.

==============================
This study was performed to evaluate the effects of rye silage treated with sodium formate (Na-Fa) and lactic acid bacteria (LAB) inoculants on the ruminal fermentation characteristics, methane yield and energy balance in Hanwoo steers. Forage rye was harvested in May 2019 and ensiled without additives (control) or with either a LAB inoculant or Na-Fa. The LAB (Lactobacillus plantarum) were inoculated at 1.5 × 1010 CFU/g fresh matter, and the inoculant was sprayed onto the forage rye during wrapping at a rate of 4 L/ton of fresh rye forage. Sixteen percent of the Na-Fa solution was sprayed at a rate of approximately 6.6 L/ton. Hanwoo steers (body weight 275 ± 8.4 kg (n = 3, group 1); average body weight 360 ± 32.1 kg (n = 3, group 2)) were allocated into two pens equipped with individual feeding gates and used in duplicated 3 × 3 Latin square design. The experimental diet was fed twice daily (09:00 and 18:00) during the experimental period. Each period comprised 10 days for adaptation to the pen and 9 days for measurements in a direct respiratory chamber. The body weights of the steers were measured at the beginning and at the end of the experiment. Feces and urine were collected for 5 days after 1 day of adaptation to the chamber, methane production was measured for 2 days, and ruminal fluid was collected on the final day. In the LAB group, the ratio of acetic acid in the rumen fluid was significantly lower (p = 0.044) and the ratio of propionic acid in the rumen fluid was significantly higher (p = 0.017). Methane production per DDMI of the Na-FA treatment group was lower than that of the other groups (p = 0.052), and methane production per DNDFI of the LAB treatment group was higher than that of the other groups (p = 0.056). The use of an acid-based additive in silage production has a positive effect on net energy and has the potential to reduce enteric methane emissions in ruminants.

Introduction

Because interest in the environment has increased owing to the climate crisis, greenhouse gases released by the livestock industry have been identified as a problem that requires solving. Consequently, the amount of methane (CH4) emissions from various livestock animals has been studied worldwide, and the Intergovernmental Panel on Climate Change (IPCC) has provided official guidelines to estimate the amount of CH4 in animals. In South Korea, where cold winters can last for 4 months or longer and cattle are raised in intensive systems, successful forage conservation is crucial for providing high nutritive value feed and ensuring potential self-sustaining forage in the livestock industry. Wrapped round bale silage, fermented by spontaneously existing lactic acid bacteria (LAB) under anaerobic conditions, has mainly been used to conserve forage, such as Italian Ryegrass, Rye, and whole-crop rice, because it is considered relatively easy to use. In silage, the pH drop is generally caused by the production of organic acids through extensive fermentation of inoculated LAB (Cushnahan & Mayne, 1995; Huhtanen, Brotz & Satter, 1997), and thus has the advantage of the lowest loss of dry matter (DM) and energy from the forage (Borreani et al., 2018) during storage when compared to clostridial fermentation, which causes high butyric acid production (Driehuis et al., 2018). In South Korea, total mixed ration (TMR) is the most frequently used feed for ruminants, and commercial LAB have generally been inoculated to enhance the preservation period. Acid-based additives, such as formic acid and sodium formate (Na-FA), have become commercial silage additives because they rapidly reduce the pH in the early phase of fermentation and inhibit the growth of spoilage bacteria (Muck et al., 2018) while increasing the lactic acid concentration (Cussen et al., 1995). Treatment of alfalfa silage with formic acid positively affects nitrogen utilization in dairy cows (Broderick et al., 2007). Therefore, acid-based additives along with LAB inoculants have been widely used in the livestock industry to produce silage.

It has been suggested that increasing the inoculant-enzyme in silage may reduce enteric CH4 emissions due to increased propionate production in the rumen (Shingfield, Jaakkola & Huhtanen, 2002) compared to the direct addition of formic acid (Boadi et al., 2004). However, silage inoculated with LAB did not show any changes in fermentation characteristics, and there was no change in CH4 emissions compared with untreated silage in a feeding trial of dairy cows (Ellis et al., 2016). Many studies have estimated enteric CH4 in livestock animals; however, there is limited research regarding the effect of silage fermentation characteristics on CH4 emissions of ruminants (Hristov et al., 2013; Knapp et al., 2014; Doyle et al., 2019). Although some studies have investigated the effect of LAB-treated silage on enteric CH4 emission in ruminants, studies on the effects of acid-based additives on enteric CH4 emission are rarer than those on LAB in ruminants.

The temperate zones of East Asia, including South Korea, often face challenges during silage production because forage is harvested during the rainy season (Daniel et al., 2019). Wet forage is more difficult to preserve than dry forage because spoilage bacteria grow faster under high moisture conditions (Daniel et al., 2019). The use of acid-based materials in silage production could improve the safety of silage preservation. However, to confirm whether the use of acid-based additives decreases or increases enteric CH4 emissions from the livestock industry, it is essential to evaluate enteric CH4 emissions when ruminants are fed acid-based additive-treated silage.

Therefore, this study aimed to assess the effect of treating rye silage with Na-Fa and LAB inoculants on the ruminal fermentation characteristics and CH4 yield of Hanwoo steers.

Materials and Methods

Ensiling properties of rye (Experiment 1)

Silage preparation

The forage rye was harvested using a rotary mower (Frontier RC2048, John Deere, IL, USA) in the morning of May 2019 and ensiled by noon on the same day using a non-chopper round baler (460M, John Deere) with no additives (control), either with LAB inoculant or Na-Fa (Fig. 1). The LAB inoculant (Lactobacillus plantarum 1.5 × 1010 CFU/g fresh matter; CM Bio, Anseong, South Korea) was diluted to a concentration of g/L of tap water and sprayed on the forage rye during the wrapping at a rate of approximately 4 L/ton of fresh forage rye. Sixteen percent of the Na-Fa solution was sprayed at a rate of approximately 6.6 L/ton. The optimum baler running speed was determined on the pre-manufacturing day to ensure the planned spraying amount of the additive solution. The finished bales were wrapped in 15 layers of film and stacked for 60 days at a farm at Seoul National University (latitude: 37.5202213; longitude: 128.4439284; 1447-1, Pyeongchang-daero, Pyeongchang-gun, South Korea). Prior to bale silage preparation for each treatment, forage rye was collected at approximately 10 m intervals for chemical analysis. More than four bale silages were selected from each treatment after 8 weeks of fermentation, and nine core samples were obtained from each bale silage using a sampler (Push Type t-handle Uni-forage Sampler, Star Quality Samplers Inc., AB, Canada). The core samples were pooled to produce one sample per bale, and the samples were frozen until analysis of fermentation characteristics and chemical composition.

Figure 1 Picture of rye silage, lactic acid bacteria treated rye silage, and sodium formate treated rye silage used in this experiment.

(A) Rye silage (control); (B) lactic acid bacteria treated rye silage; (C) sodium formate treated rye silage.

Fermentation profiles and chemical analysis

Silage samples were dried for 72 h in a convection dry oven (model VS-1202D9, Vision Scientific, Daejeon-Si, South Korea) at 65 °C, and the DM content was calculated. The dried samples were ground and passed through a 1 mm sieve (Thomas Scientific Model 4, Swedesboro, NJ, USA) and used to analyze the nitrogen (AOAC method 990.0), crude ash (AOAC method 942.05), and ether extract (EE; AOAC method 920.39) contents. Neutral detergent fiber (NDF) and acid detergent fiber (ADF) contents were estimated using the methods of Van Soest, Robertson & Lewis (1991) and Van Soest (1973), respectively. Gross energy (GE) was determined using an automatic bomb calorimeter (Parr, 1261 bomb calorimeter, Parr Instruments Co., Moline, IL, USA).

The raw silage sample (10 g) was placed in 100 mL of distilled water and stored overnight, after which the extract was passed through a filter paper (Whatman No. 6, Sigma-Aldrich, St. Louis, MO, USA). The pH (pH meter, Mettler-Toledo, Schwerzenbach, Switzerland) was immediately measured, and the extract was stored at −20 °C until the analysis of NH4-N (Broderick & Kang, 1980), water-soluble carbohydrate (WSC) (Yemm & Willis, 1954), and lactic acid using an assay kit (Megazyme, Bray Co., Wicklow, Ireland). The silage extract was centrifuged at 1,500× g for 15 min at 4 °C (T04B, Hanil Science Industrial, Daejeon, South Korea) and pretreated using the method described by (Erwin, Marco & Emery, 1961) for volatile fatty acid (VFA) analysis. A gas chromatograph equipped with a flame ionization detector and an FFAP CB column (25 m × 0.32 mm, 0.3 μm; Agilent Technologies, Santa Clara, CA, USA) was used to measure the VFA contents. The NH4-N concentration was measured using a modified colorimetric method (Chaney & Marbach, 1962).

Nutrients digestibility, rumen fermentation characteries, and CH4 production (Experiment 2)

Animals, experimental design, and diet

The experimental animals (Hanwoo) were obtained from commercial cattle auctions in South Korea. The cattle were housed in feedlots (W × L, 5 × 10 m) per four cattle at the Seoul National University farm (latitude, 37.5202213; longitude, 128.4439284; 1447-1, Pyeongchang-daero, Pyeongchang-gun, South Korea) until experimentation. The experimental feed (Table 1) was provided once daily at 10:00, and water was provided ad libitum. The feed consisted of Timothy hay and commercial concentrate feed until the end of the experiment. The cattle were raised on the same farm until slaughter at the end of the experiment. In vivo experiments were conducted with the approval of the Seoul National University Institutional Animal Care and Use Committee (approval number: SNU-181127-12).

Table 1 Chemical compositions of rye silage, lactic acid bacteria treated rye silage, and sodium formate treated rye silage.

Item	Additives1 (n = 6)	
Control	LAB	Na-FA	
Ingredients2, %DM				
Rye silage	93.41			
Rye silage with LAB		93.41		
Rye silage with Na-FA			93.41	
Concentrates3	6.59	6.59	6.59	
Total	100.0	100.0	100.0	
Notes:

DM, dry matter; NDF, neutral detergent fiber; ADF, acid detergent fiber; WSC, water soluble carbohydrate GE, gross energy.

1 Control, no treated rye silage; LAB, rye silage treated with lactic acid bacteria; Na-FA, rye silage treated with sodium formate.

2 Mineral block and water were fed ad-libitum.

3 Concentrate is used commercial product included NDF contents. Crude protein 17.0%; NDF, 18.0%; NFC, 29.0%; TDN, 70%.

Growing Hanwoo steers groups 1 (average body weight 275 ± 8.4 kg, n = 3) and 2 (average body weight 360 ± 32.1 kg, n =3) were randomly allocated into two adjacent pens with a duplicated 3 × 3 Latin square design, and the experimental unit was each animal. In each pen equipped with individual feeding gates, three steers individually consumed 2.7% of their body weight on a feed basis of different silages (control, LAB, and Na-Fa) and 0.2% of concentrate in the morning (09:00) and evening (18:00). The ingredient composition of the concentrate was DM = 3.19%, crude protein (CP) = 16.69% of DM, EE = 5.0% of DM, ash = 6.93% of DM, and crude fiber = 4.09% of DM. Bale silage was chopped using a TMR mixer for 5 min, packed into four or five bags, and stored at room temperature. Each period comprised 10 days of adaptation to the pen and 9 days of measurement in a direct respiratory chamber. Steers had free access to water and mineral blocks throughout the experiment. The body weights of the steers were measured at the beginning and end of the experiment. On the same day, when group 1 returned to their pen after completing the measurements in the chamber, group 2 moved to the chambers because the steers had already completed feed adaptation in the pen. After 1 day of adaptation in a chamber, feces and urine were collected for 5 days, CH4 production was measured for 2 days, and rumen fluid was collected on day 19. All in vivo experiments were performed in accordance with IPCC guidelines.

Sampling and data collection

The silage and concentrates offered for each treatment were sampled, bulked for each period, and stored in a cool room for DM measurement and composition analysis. The total weight of feces excreted daily on a polyvinyl chloride sheet spread on the floor of each chamber was collected and weighed at 09:00 daily for 5 days for each period. Ten percent of homogenized daily feces from each animal was sampled and stored at −20 °C until the formation of composite samples by the steer within each period to calculate DM digestibility (DMD), organic matter (OM) digestibility, CP digestibility, and NDF digestibility. Urine was separated using a rubber pyramidal funnel to prevent urine contamination of the feces. When the urine was excreted, it was collected in a container, because the funnel was connected to a rotary vane vacuum pump using polypropylene tubing. Rumen fluid was collected from the rumen before morning feeding and 1.5 and 3.0 h after feeding and squeezed through four layers of cheesecloth to measure ruminal pH and analyze VFA and ammonia nitrogen (NH3-N) concentrations.

CH4 sampling methods and respiratory-metabolic chamber

CH4 production from enteric fermentation by steers fed the three silages was measured using three indirect open-circuit respiratory chambers. Each chamber (Fig. 2, outside dimension: 137 cm wide × 356 cm deep × 200 cm tall) made of steel frames had a feeder and a water bowl and maintained 18–25 °C and 50–60% relative humidity by an independent regulatory system that contained an air conditioner (Dryer DK-150E, Hangzhou Duokai Technology Co., Hangzhou, China) with a recirculating fan and air filter (model ALFFIZ-WBCAI-015H, Busung, India). For recirculation, air exited the chamber (0.6 m/s) through openings in the middle of the chamber ceiling, and the dehumidified air was recycled into the chamber (0.1 m/s) through openings at the back end of the chamber ceiling. The air circulation speed was fixed to avoid positive pressure inside the chamber and to achieve maximum gas recovery. The condensed water from the refrigeration unit was drained into the chamber through a capillary tube attached to the water traps.

Figure 2 Picture of respiratory-metabolic chamber and gas analyzer in this experiment.

(A) Whole respiratory-metabolic chamber; (B) methane, carbon dioxide, and oxygen gas analyzer.

Each chamber was maintained at a negative pressure using a sealed rotary exhaust pump connected to a flow meter (model LS-3D, Teledyne Technologies, Thousand Oaks, CA, USA) that provided a constant wet ventilation rate of 450 L/min. The air outflow from each chamber was subsampled from the PVC pipe between the flow meter and exhaust pump using the gas sampling pump of the analysis system. All three chambers shared a common gas analysis system that comprised a multiplexer (Metabolic controller, B.S. Technolab, Seoul, South Korea) connected to a gas sampling pump (Columbus Instruments, Columbus, OH, USA) and analyzers (model VA-3000, Horiba Ltd., Kyoto, Japan) containing nondispersive infrared CH4 (0–2,000 ppm). The air was subsampled every 14 min with a sequence of background air and chamber air sampling of 90 and 120 s for each chamber. The average value at the final sampling time of 20 s was used for the calculations. The recovery rate in each chamber was tested at the beginning of the experiment using a standard CH4 gas mixture (25% mol/mol balance N2; Air Korea, Seoul, South Korea). CH4 sampling was performed according to the IPCC guidelines.

Chemical composition analysis

Fecal samples were dried for 72 h in a convection dry oven (model VS-1202D9, Vision Scientific, New Milford, CT, USA) at 65 °C, and the DM content was calculated. The dried samples were ground to pass through a 1 mm sieve (Thomas Scientific Model 4) and used to analyze the nitrogen (AOAC method 990.0), crude ash (AOAC method 942.05), and EE (AOAC method 920.39) contents. NDF and ADF contents were estimated using the methods of Van Soest, Robertson & Lewis (1991) and Van Soest (1973), respectively.

Rumen gastric juice was pretreated using the method described by Erwin, Marco & Emery (1961) for VFA analysis. A gas chromatograph equipped with a flame ionization detector and an FFAP CB column (25 m × 0.32 mm, 0.3 μm; Agilent Technologies, Santa Clara, CA, USA) was used to measure the VFA contents. The NH3-N concentration was measured using a modified colorimetric method (Chaney & Marbach, 1962).

Statistical analysis

Data were analyzed using PROC MIXED in SAS (Version 9.4; SAS Institute Inc., Cary, NC, USA). The model considered diet as the fixed effect and the two animals and periods as random effects. Outliers were removed according to the method of interquartile range and, if not outliers, all data were included. If the effect of double blocking was not significant, the significance among treatments was evaluated such that the model did not include both animals and periods as block factors. Significance of the treatments were contrasted using the PDIFF option. Treatment effects were considered significant at p < 0.05, and trends were evaluated at 0.05 ≤ p < 0.10. All data were presented as least-squares means.

Results and discussion

Silage fermentation characteristics (Experiment 1)

The chemical compositions of the silages used in this study are listed in Table 2. The DM content did not differ significantly among the treatments. The OM content was highest in the LAB group and lowest in the control group (p = 0.046). The CP content of the Na-FA group was the highest among all treatments and that of the control group was the lowest among all treatments (p = 0.036). The EE content did not differ significantly among the treatments. The NDF content of the control group was the highest among all treatments, whereas that of the Na-FA group was the lowest (p = 0.018). The ADF content of the silages did not differ significantly among treatments. The WSC content in the Na-FA group was higher than that in the other groups (p = 0.039).

Table 2 Chemical compositions of rye silage, lactic acid bacteria treated rye silage, and sodium formate treated rye silage.

Item	Additives1 (n = 6)	SEM	p-value	
Control	LAB	Na-FA	
DM, %	22.4	22.2	22.7	0.094	0.467	
OM, % DM	92.5c	93.3a	92.8b	0.088	0.046	
CP, % DM	10.9b	11.1b	11.9a	0.124	0.036	
EE, % DM	2.7	2.9	3.2	0.084	0.104	
NDF, % DM	57.6a	56.4b	54.8c	0.272	0.018	
ADF, % DM	38.4	35.5	35.9	0.517	0.232	
WSC, % DM	3.2b	3.4b	4.4a	0.147	0.039	
Gross energy, kcal/kg DM	4,440.3	4,473.5	4,532.2	33.4	0.288	
Notes:

SEM, standard error of the means; DM, dry matter; OM, organic matter; CP, crude protein; EE, ether extract; NDF, neutral detergent fiber; ADF, acid detergent fiber; WSC, water soluble carbohydrate.

1 Control, no treated rye silage; LAB, rye silage treated with lactic acid bacteria; Na-FA, rye silage treated with sodium formate.

abcMeans in the same raw with different superscripts differ significantly p < 0.05.

A previous study indicated that the control group had a greater number of microbial groups than silage during fermentation (Franco et al., 2022). Thus, the addition of acid-based additives during silage production may restrict microbial diversity in silages. Furthermore, the use of acids to lower the pH of feed can help prevent spoilage and harmful fungi (Buerman, Worobo & Padilla-Zakour, 2020). The changes in CP, WSC, and OM contents during silage fermentation observed in this study were similar to those reported in a previous study (Franco et al., 2022). The addition of LAB and acidic additives in silage had an effect on the composition of OM (CP, NDF, and WSC), as observed in both studies. However, a difference in the WSC content was observed between the formic acid groups (Na-FA group in this study) and the LAB and control groups in both the present study and a previous experiment (Franco et al., 2022). Addition of acid during silage production increased the level of soluble sugars. Typically, to achieve a decrease in pH during silage production, microorganisms must grow and produce sufficient fatty acids (Lund et al., 2020). Directly adding acid-based additives to silage to reduce the pH does not require microbial growth to produce fatty acids, which can prevent unnecessary consumption of nutrients in forage feed. Therefore, directly adding fatty acids is considered a better method than using microbial inoculants to conserve feed nutrients during silage production.

Silage fermentation characteristics, as shown in Table 3 and Fig. 3, indicate that the pH of silages is significantly higher in the control group and significantly lower in the LAB group (p < 0.01). The acetic and butyric acid contents were significantly higher in the control group than in the other groups (p < 0.01). Lactic acid content was significantly higher in the LAB and Na-FA groups than in the control group (p < 0.01). The NH3-N content of the silages was significantly higher in the control group and significantly lower in the Na-FA group (p < 0.01).

Table 3 Silage fermentation characteristics of rye silage, lactic acid bacteria treated rye silage, and sodium formate treated rye silage.

Item	Additives1 (n = 6)	SEM	p-value	
Control	LAB	Na-FA	
pH	4.8a	3.8c	4.1b	0.040	<0.001	
Total VFA, % DM						
Acetic acid	1.87a	0.69b	0.73b	0.040	<0.001	
Butyric acid	1.81a	0.23b	0.36b	0.085	<0.001	
Lactic acid	0.53b	3.02a	2.86a	0.064	<0.001	
NH4-N, % DM	15.6a	12.8b	12.0c	0.215	<0.001	
Notes:

SEM, standard error of the means; VFA, volatile fatty acid; NH4-N, ammonia nitrogen.

1 Control, no treated rye silage; LAB, rye silage treated with lactic acid bacteria; Na-FA, rye silage treated with sodium formate.

abcMeans in the same raw with different superscripts differ significantly p < 0.05.

Figure 3 Silage fermentation characteristics rye silage, lactic acid bacteria treated Rye silage, and sodium formate treated rye silage.

(A) pH; (B) lactic acid; (C) butyric acid; (D) ammonia nitrogen; Control, no treated rye silage; LAB, rye silage treated with lactic acid bacteria; Na-FA, rye silage treated with sodium formate. Error bar means standard error of means and abcMeans in the same raw with different superscripts differ significantly p < 0.05.

Silage is commonly used for storing forage to minimize feed loss and allow long-term storage in the ruminant industry. The success of silage production depends on achieving a decrease in pH, which is largely caused by the growth of microorganisms and the production of secondary metabolites, such as fatty acids, organic acids, and NH3-N. (Doelle, 2014). Generally, the growth of spoilage bacteria is inhibited under pH conditions below 5.5 (Gill, 2003). In this study, the pH values of the silages (control, LAB, and Na-FA groups) were sufficiently low (ranging from 4.1 to 4.8) to prevent spoilage. VFAs are the primary secondary metabolites produced by microbial activity during fermentation. Acetic acid bacteria produce acetic acid as a secondary metabolite after utilizing carbon and nitrogen for microbial growth (Guillamón & Mas, 2011). Acetic acid can be found in various fermented products and acts as a mild antibacterial agent by decreasing silage pH. LAB are classified into two types via “Homolactic” and “Heterolactic” metabolism, which produce various VFAs including lactate and acetate (Gänzle, 2015). Lactate and acetate are mainly produced using pyruvate and acetyl phosphate, respectively, and acetate is also used in lactate metabolism to produce lactate by LAB (Gänzle, 2015). This metabolism leads to lower acetic acid content in the LAB group than in the control group (Table 3). In another study, LAB strains were observed without LAB inoculation in silage, which showed that inoculation with formic acid and LAB led to an increase in lactic acid content, and that the relative abundance of the LAB strain increased with the addition of formic acid and LAB inoculation (Lei et al., 2023). Furthermore, in the group to which formic acid and LAB were added, the acetic acid content increased by day 15 and subsequently decreased by day 45 (Lei et al., 2023). LAB grow more robustly at pH values of 5 than at pH 3; however, low pH conditions are favorable for acid purification because of the high acid resistance of LAB (Lund et al., 2020). A decrease in acetic acid after using the acidic additive can be explained in two ways: 1) the dominance of LAB by high acid resistance compared to other microbes, and 2) acetate used to produce lactic acid by LAB metabolism. Butyric acid is mainly produced by anaerobic bacteria such as Clostridium sp. and is considered a poor indicator of fermentation quality. Although it can lower the pH of fermented products, butyric acid bacteria can interrupt the decrease in pH by producing butanol (Xue & Cheng, 2019). Additionally, in the rumen, butyric acid causes health problems that increase the risk of ketosis (Vicente et al., 2014). The decrease in pH during silage production can be attributed to the production of various fatty and organic acids by microbes, even in the absence of additives. However, this process carries risks such as spoilage, abnormal fermentation, and the presence of unexpected bacteria. It is important to determine whether one can control the risks associated with silage production. In this study, the pH of the control group was higher than that of the treatment groups, which appeared to be caused by butyric acid fermentation (Table 3). This suggests that the decreased in pH during silage production can be controlled without the risk of unexpected fermentation using acid-based additives or LAB. The addition of LAB creates an environment where Lactobacillus species dominate in silage, and these bacteria consume nutrients from the forage to grow and produce various acids, resulting in a decrease in pH (Moreno & Peinado, 2012). The growth of Lactobacillus species in silage can reduce the nutrients available in feed, resulting in a decrease in the nutrients available for livestock animals to utilize. In contrast, Na-FA addition can decrease pH without microbial growth by directly supplying an acid. In other words, Na-FA, which adds acids directly to silage, has an advantage over LAB additives because it may reduce nutrient consumption by microbes in forage. This can be explained by the higher OM content in the LAB group than in the Na-FA group; however, the GE of the Na-FA group was higher than that of the LAB group (Table 2). Treating forage with acid before silage production increases the acid hydrolysis of NDF, which can lead to the production of more sugars (McDonald, Henderson & Heron, 1991). Clearly, the addition of LAB and Na-FA during silage production can help conserve the nutrients fed to animals in silage. The choice of additive can be based on whether there is more benefit in preserving the WSC or NDF content in the silage.

NH3-N is a secondary metabolite that is produced as microorganisms grow during fermentation. The NH3-N in silage is associated with reduced silage intake, predominantly as a product of clostridial fermentation of amino acids (Charmley, 2001). A low pH can generally help suppress NH3-N production during fermentation (Oladosu et al., 2016). Although the NH3-N content of the Na-FA group was lower than that of the LAB group in this study, both Na-FA and LAB additives were effective in reducing NH3-N production by inhibiting microbial growth compared to the control group.

Nutrients digestibility, rumen fermentation characteries, and CH4 production (Experiment 2)

Rumen fermentation characteristics

The rumen fermentation characteristics of rye silage, LAB-treated rye silage, and Na-Fa-treated rye silage in Hanwoo cattle are shown in Table 4. The pH and total VFA content of the rumen fluid did not differ among the treatment groups. The ratio of acetic acid in the control, LAB, and Na-Fa groups were 69.9, 67.5, and 68.5% mol, respectively (p = 0.044). The ratio of propionic acid in the control, LAB, and Na-Fa groups were 17.0, 19.6, and 18.4% mol, respectively (p = 0.017). Butyric, iso-butyric, and valeric acid contents of the rumen fluid did not differ among the treatment groups. The ratios of isovaleric acid in the control, LAB, and Na-Fa groups were 1.23, 1.39 and 1.58% mol, respectively (p = 0.004). The NH3-N content of the rumen fluid did not differ among the treatment groups.

Table 4 Effects of lactic acid bacteria and sodium formate treated rye silage on ruminal fermentation characteristics of Hanwoo.

Item	Additives1 (n = 6)	SEM	p-value	
Control	LAB	Na-FA	
pH	6.99	6.95	7.06	0.12	0.824	
Total VFA, mol	67.1	58.7	63.4	4.73	0.492	
Acetic acid, % mol	69.9a	67.5b	68.5ab	0.78	0.044	
Propionic acid, % mol	17.0b	19.6a	18.4ab	0.62	0.017	
Butyric acid, % mol	9.60	8.78	8.86	0.39	0.279	
Iso-butyric acid, % mol	0.98	1.12	1.19	0.06	0.061	
Valeric acid, % mol	1.28	1.52	1.44	0.11	0.071	
Iso-valeric acid, % mol	1.23b	1.39ab	1.58a	0.09	0.004	
NH4-N, mg/dL	10.1	9.9	11.7	1.73	0.655	
Notes:

VFA, volatile fatty acid; NH4-N, ammonia nitrogen; SEM, standard error of the means.

1 Control, additives-free rye forage silage group; LAB, rye forage silage group treated with lactic acid bacteria; Na-FA, rye forage silage group treated with sodium format.

abMeans in the same raw with different superscripts differ significantly p < 0.05.

Rumen contains a complex microbiota comprising bacteria, ciliated protozoa, archaeal fungi, and viruses that degrade a broad range of feed ingredients (Firkins & Yu, 2015). A large population of microbiota in the rumen protects against various shocks caused by feed ingredients. However, when large amounts of rapidly degradable ingredients are supplied to the rumen, its pH decreases, leading to a condition known as subacute ruminal acidosis (SARA) (Kleen et al., 2003). SARA is typically diagnosed when pH drops below 5.5 (Kleen et al., 2003). This study showed pH values of 6.99–7.06, and therefore, it is considered that the silage had little effect on decreases in pH and did not lead to SARA. In the present study, the control group had the highest levels of acetic acid in the rumen (Table 4). This difference in acetic acid production could be attributed to variations in the acetic acid and NDF contents of the experimental feed, because the production of acetic acid in the rumen is dependent on the NDF content of the forage (Sutton et al., 2003). In the rumen, propionate is primarily produced by propionic acid bacteria such as Selenomonas, Ruminobacter, Prevotella, and Clostridium through the succinic pathway. This pathway involves key enzymes such as propionyl CoA carboxylase and succinate CoA synthetase (Wang et al., 2020). The succinate pathway is initiated by providing pyruvate (a three-carbon molecule) derived from glucose (a six-carbon sugar). Hexose in cattle feed contains WSC and non-fiber carbohydrate fractions (Yang, 2005). Lactic acid is an important precursor of VFAs in the rumen, and there have been reports that lactic acid changes to acetic acid and propionate approximately 40% and 32% of the time, respectively (Gill et al., 1986; Nakamura & Takahashi, 1971). In rumen, butyric acid converts 93% of acetic acid or acetic acid freely converts to butyric acid (Gill et al., 1984). In the present study, the elevated acetic acid content in the rumen of the control group could be attributed to the provision of butyric acid in the diet. Conversely, it is evident that the increased propionate content in the rumen was a result of lactic acid originating from the experimental feed.

Isoacids are synthesized mainly as metabolites by the degradation of amino acids, such as valine, isoleucine, leucine, and proline, which are converted into amino acids and branched fatty acids by rumen microbes (Andries et al., 1987). Because the CP content of the LAB and Na-FA groups in the silage was greater than that of the control group (Table 4), the production of isoacids in the rumen was considered to have been influenced. Additionally, iso-butyric and valeric acids in the LAB and Na-FA groups tended to be higher than those in the control group (Table 4), which could be attributed to the higher protein intake of the LAB and Na-FA groups compared to the control group (Table 5). In this study, LAB and acid treatment resulted in an increase in silage protein content, which is thought to elevate isoacids (iso-butyric and iso-valeric acids) in rumen fluid when fed to ruminants. Therefore, feeding silage to ruminants may increase the amount of isoacids in the rumen, which are specific nutrients for ruminal cellulolytic bacteria, and appear to positively affect rumen microbial fermentation (Fievez et al., 2012). Consequently, feeding silage to ruminants might result in a weak increase in nutrient digestibility in the rumen.

Table 5 Effects of lactic acid bacteria and sodium formate treated rye silage on nutrient digestibility of Hanwoo.

Items	Additives1 (n = 6)	SEM	p-value	
Control	LAB	Na-FA	
Intake, kg/d						
DMI	4.59	4.56	4.58	0.21	0.994	
Digestibility, %						
DMD	69.5	73.8	71.6	2.53	0.168	
OMD	70.7	75.1	73.0	2.58	0.157	
CPD	70.3	73.2	72.4	2.47	0.411	
NDFD	69.1	72.1	69.5	2.83	0.388	
Notes:

SEM, standard error of the mean; DMD, dry matter digestibility; OMD, organic matter digestibility; CPD, crude protein digestibility; NDFD, neutral detergent fiber digestibility.

1 Control, untreated rye silage; LAB, rye silage treated with lactic acid bacteria; Na-FA, rye silage treated with sodium formate.

Nutrients digestibility

Nutrient digestibility of rye silage, LAB-treated rye silage, and Na-Fa-treated rye silage by Hanwoo cattle is shown in Table 5. The intake of DM, CP, and NDF did not differ among the treatment groups, and the digestibility of DM, OM, CP, and NDF did not differ among treatment groups.

Silage is preserved by acidification via microbial fermentation, and various microorganisms consume nutrients for microbial growth during silage production (Muck et al., 2018). It decreases the DM and OM contents of forage, and carbon sources can be converted and excreted as carbon dioxide and fatty acids. Thus, silage production can result in a decrease in nutrients in the forage based on DM. Moreover, long-term storage of silage can lead to increased costs owing to nutrient loss. Although the DM content of forage can decrease during silage production owing to the conversion of nutrients to fatty acids, fatty acids have some advantages, such as an increased digestibility rate of feed compared to non-silage forage in the rumen (Ferraretto, Shaver & Lauer, 2014). Ruminants absorb and utilize VFAs produced by microorganisms as nutrients in the rumen. VFAs produced by microbes during silage production can be utilized by being absorbed directly through the inner rumen-wall cells (Bedford et al., 2020). In this study, the digestibility of all the nutrient fractions in the experimental diets did not differ significantly among the treatments. Although feeding trials and sampling were designed to reduce experimental errors, the complex nature of ruminal gut digestion makes it difficult to fully explain this situation. Several studies have reported improvements in DM and NDF digestibility per unit of forage feed with silage (Miron et al., 2007; Cotanch et al., 2012). Although silage has been shown to improve the digestibility of DM and NDF per unit of forage feed, the DM intake of silage forage is often lower than that of non-silage forage (Charmley, 2001). This means that although the digestibility per unit of forage may be higher, the overall feed intake may be lower, potentially resulting in a lower overall nutrient intake. Therefore, it is important to carefully consider the trade-offs between increased digestibility and potentially lower feed intake when silage is used as feed for ruminants.

Energy balance

The effects of LAB and Na-Fa treatments on the energy balance of Hanwoo cattle are shown in Table 6. GE intake did not differ among the treatment groups. Energy loss from feces, urine, and CH4 did not differ among the treatment groups. Energy loss from heat was significantly greater in the LAB group than in the Na-FA group (p = 0.045). The ratios of energy loss from feces, urine, and CH4 did not differ among the treatment groups. The ratio of energy loss from heat was significantly lower in the Na-FA group than that in the other groups (p = 0.049). Digestible energy (DE) and metabolizable energy (ME) did not differ among the treatment groups. The net energy (NE) of the LAB-treated group was significantly higher than those of the other groups (p = 0.010). The proportion of DE and ME (% of GE basis) did not differ among the treatment groups, and NE (% of GE basis) was significantly lower in the Na-FA group than in the other groups (p = 0.009).

Table 6 Effects of lactic acid bacteria and sodium formate treated rye silage on energy balance of Hanwoo.

	Additives1 (n = 6)			
Item	Control	LAB	Na-FA	SEM	p-value	
Gross energy intake, Mcal/d	20.3	20.4	20.8	0.84	0.939	
Energy loss, Mcal/d						
Feces	6.16	5.36	5.89	0.37	0.183	
Urine	2.28	2.79	2.55	0.26	0.386	
Methane	1.50	1.57	1.60	0.15	0.650	
Heat	8.62ab	9.29a	7.99b	0.44	0.045	
Energy loss, %GE						
Feces	30.7	26.4	28.3	2.73	0.387	
Urine	11.2	13.8	12.3	1.91	0.363	
Methane	7.42	7.69	7.71	1.07	0.992	
Heat	42.4a	45.4a	38.4b	2.07	0.049	
Digestible energy, Mcal/d	14.1	15.0	14.9	0.67	0.083	
Metabolizable energy, Mcal/d	10.3	10.7	10.8	0.57	0.267	
Net energy, Mcal/d	1.74ab	1.29b	2.27a	0.56	0.010	
Proportion, %GE						
Digestible energy	69.5	73.5	71.6	2.56	0.387	
Metabolizable energy	50.7	52.5	51.9	0.21	0.583	
Net energy	8.58b	6.35c	11.18a	0.32	0.009	
Notes:

SEM, standard error of the means; GE, gross energy.

1 Control, untreated rye silage; LAB, rye silage treated with lactic acid bacteria; Na-FA, rye silage treated with sodium formate.

abcMeans in the same raw with different superscripts differ significantly p < 0.05.

The GE of the feed was measured as the energy required for complete combustion with oxygen, using a bomb calorimeter (AOAC, 2005). However, this does not indicate the total energy available in the feed. Generally, in animals, a portion of the energy in feed is released in various forms, such as feces, urine, gases, and heat, during digestion (National Research Council, 2016). The ME value is determined by subtracting the urine and gas energy from the DE value; the gas energy excreted by ruminants is mainly CH4 (National Research Council, 2016). In this study, energy loss values from feces, urine, and CH4, including DE and ME values calculated using the energy loss values, did not differ among the treatment groups. However, the DE tended to be higher in the LAB and Na-FA group than in the control group (p = 0.083). The heat energy loss of the LAB group was significantly greater than that of the other treatment groups, and that of the Na-FA group was the lowest among all the treatment groups (Table 6, p = 0.045 and p = 0.049, respectively). The NE of the LAB group was the lowest among all treatment groups, and that of the Na-FA groups was the lowest among all treatment groups (Table 6; p = 0.010 and p = 0.009, respectively). The change in heat energy is a complex concept that is difficult to accurately determine in animals. Thus, in live animals, heat increment is generally determined by subtracting the heat loss of fasting animals from that of eating animals (Cherian, 2019). Heat increase occurs due to the act of eating, chewing, digesting feed, and absorbing nutrients from the gut in livestock animals, assuming that they live in similar environments. Thus, the difference in heat loss in this study was considered to be caused by the effect of the additives. In recent studies in which silage inoculated with LAB was fed to ruminants, changes in microbial communities were influenced by changes in rumen metabolites (Han et al., 2022). Heat production in the rumen is caused by glucose accumulation, which occurs after an increase in glucose concentration in the rumen (Russell, 1986). Lactic acid is synthesized as the end metabolite of glucose metabolism and is utilized in gluconeogenesis and oxidation (Larsen & Kristensen, 2013). Thus, heat loss in the LAB group could be attributed to an increase in heat loss due to an increase in lactic acid supply.

Formic acid is the major secondary metabolite produced by various microbes in the rumen and contributes as much to CH4 production as acetate and hydrogen gas (Asanuma, Iwamoto & Hino, 1998). The direct addition of formic acid during silage production contributes to an increase in WSC and hydrolysis of the NDF fraction in the forage feed (Muck et al., 2018), which reduces the population of microbial communities vulnerable to acid. This could improve nutrient absorption in the rumen by supplying rapidly degradable carbohydrates instead of LAB. However, in the present study, the lactic acid content in the Na-FA group did not differ from that in the LAB group (Table 3). Even if the usage of additives was similar between a previous study (Franco et al., 2022) and this study, the effect of additives was not the same during silage production. The differences between both experiments are as follows: 1) Franco et al. (2022) used a mixed additive (formic acid, Na-Fa, propionic acid, and potassium sorbate), whereas this study used only 60% Na-Fa, and 2) the fermentation period differed (previous study: 93 days; this study: 60 days). The key difference between formic acid and Na-Fa is that the hydrogen ions are replaced by sodium ions to form salts. Formic acid is hydrophilic, readily soluble in water, and dissociates into formate ions (HCOO-) and hydrogen ions (H+) (Wu et al., 2004). The formate ion passes freely through the cell walls of bacteria, destroys the DNA of microbes, and the remaining hydrogen ions lead to further acidification (Lastauskienė et al., 2014). Na-Fa is hydrophilic, readily soluble in water, and dissociates into formate ions (HCOO−) and sodium ions (Na+) in water. Na-Fa does not release any hydrogen ions and does not have an acidifying ability. However, Na-Fa can prevent pathogenic microbial growth owing to dissociated formate ions (Lastauskienė et al., 2014). In this study, there was no sustained acidification due to the absence of hydrogen ions, and the acidification of the Na-FA group was considered to be caused by lactic acid from LAB growth. Thus, the effect of Na-FA is due to the acidification of lactic acid by LAB after formate ions restrain harmful microbial growth. As microbial growth by formate ions in silage decreases, it is possible that the NE for animals in the Na-FA group could be maintained compared to the other treatment groups.

CH4 production

The effects of LAB- and Na-Fa-treated rye silages on CH4 production in Hanwoo cattle are shown in Table 7. Daily CH4 production, CH4 production per DMI, CH4 production per DDMI, CH4 production per NDFI, CH4 production per DNDFI, and the CH4 conversion factor (Ym) did not differ among the treatment groups. CH4 production per DDMI of the Na-FA treatment group was lower than that of the other groups (p = 0.052), and CH4 production per DNDFI of the LAB treatment group was higher than that of the other groups (p = 0.056).

Table 7 Effects of lactic acid bacteria and sodium formate treated rye silage on methane production of Hanwoo.

Methane production	Additives1 (n = 6)	SEM	p-value	
Control	LAB	Na-FA	
CH4, g/d	63.8	68.2	61.2	3.43	0.428	
CH4, g/kg DMI	13.9	15.0	13.1	0.64	0.230	
CH4, g/kg DDMI	20.6	21.0	17.8	0.77	0.052	
CH4, g/kg NDFI	25.0	28.5	24.6	1.20	0.150	
CH4, g/kg DNDFI	37.7	41.2	34.7	1.51	0.056	
Ym	6.35	6.73	6.17	0.39	0.570	
Notes:

CH4, methane; DMI, dry matter intake; DDMI, digestible dry matter intake; NDFI, neutral detergent fiber intake; DNDFI, digestible neutral detergent fiber intake; Ym, methane conversion factor; SEM, standard error of the means.

1 Control, no treated rye silage; LAB, rye silage treated with lactic acid bacteria; Na-FA, rye silage treated with sodium formate.

In ruminants, CH4 is primarily produced by methanogenic bacteria that use acetic acid and hydrogen ions in the rumen. The high acetic acid, hydrogen, and formate contents of rumen fluid can stimulate the production of CH4 using methanogenic bacteria as precursors. In this study, lactic acid was an important precursor of VFAs in the rumen, and it can change to acetic and propionic acids approximately 40% and 32% of the time, respectively (Nakamura & Takahashi, 1971; Gill et al., 1986). In this study, the acetic acid concentration was high in the control diet, and the lactic acid concentrations in the diets of the LAB and Na-FA groups were high (Table 3). In rumen, propionic acid only converts approximately 12% to acetic acid (Gill et al., 1984). This indicates that less lactic acid can be converted to acetic acid, which may produce less CH4 by methanogenic bacteria in the rumen. Although formic acid or formate in the rumen is more readily converted to CH4 by methanogens, formate is predominantly utilized by LAB, where it is converted to lactic acid. Therefore, it is reasonable that CH4 production per DDMI and DNDFI of the Na-FA treatment group was lower than that of the other groups (Table 7). Furthermore, the key differences among the treatment groups were in the NDF and WSC contents. More acetic acid is produced by ruminal microorganisms when using forage feed than when using concentrate feed. High NDF contents produce more acetic acid in the rumen, and high concentrations contribute more to propionic acid production (Wang et al., 2020). In this study, the WSC fraction of the experimental feed in the Na-FA group was mainly a propionic acid precursor. Ruminal microbes use propionic acid to produce glucose via the gluconeogenic pathway. This indicated that the contribution of propionic acid to CH4 production was low. Thus, the addition of Na-FA could change the NDF and WSC contents during silage production, and this could affect the reduction of CH4 derived from digested DM and NDF in the experimental feed. In a previous study, formic acid-treated silage showed a dynamic change in the NDF to WSC fraction (Franco et al., 2022). It might be considered that treatment with formic acid instead of Na-Fa appeared to be more effective in reducing enteric CH4 in the rumen.

Conclusions

The results of this study revealed several differences in the effects of LAB inoculants and Na-FA on fermentation characteristics during silage production. Although the mechanism of extending the conservation period by acidification of forage is similar in LAB inoculants and Na-FA, LAB inoculants act by producing lactic acid as an acidifier, whereas Na-FA additives are directly acidified by formate ions. The direct addition of Na-FA caused a decrease in the NDF fraction and an increase in the WSC of the silage. Furthermore, this caused a difference in the VFA and fatty acid ratios between the LAB inoculants and Na-FA additives in silage. In this study, CH4 production per DDMI of the Na-FA treatment group was lower than that of the other groups, and CH4 production per DNDFI of the LAB treatment group was higher than that of the other groups. The lactic acid increase in silage by LAB addition did not affect the reduction in enteric CH4 in Hanwoo cattle. The addition of Na-FA to silage decreased enteric CH4 in Hanwoo cattle. The use of an acid-based additive in silage production positively affected the increase in NE and the potential to reduce enteric CH4 emissions in ruminants.

Supplemental Information

Supplemental Information 1 Author Checklist.

Supplemental Information 2 Raw data.

Additional Information and Declarations

Competing Interests

Author Contributions

Animal Ethics

Data Availability

Jayeon Kim is employed by Cargill Agri Purina Inc. and Geumhwi Bang is employed by Farmsco Co., Ltd. The authors declare that they have no competing interests except to Jayeon Kim and Geumhwi Bang.

Yongjun Choi conceived and designed the experiments, analyzed the data, prepared figures and/or tables, authored or reviewed drafts of the article, and approved the final draft.

Jayeon Kim performed the experiments, prepared figures and/or tables, and approved the final draft.

Geumhwi Bang performed the experiments, analyzed the data, prepared figures and/or tables, authored or reviewed drafts of the article, and approved the final draft.

Nayeon Kim performed the experiments, prepared figures and/or tables, and approved the final draft.

Krishnaraj Thirugnanasambantham performed the experiments, authored or reviewed drafts of the article, and approved the final draft.

Sangrak Lee conceived and designed the experiments, authored or reviewed drafts of the article, and approved the final draft.

Kyoung Hoon Kim conceived and designed the experiments, prepared figures and/or tables, authored or reviewed drafts of the article, and approved the final draft.

Rajaraman Bharanidharan conceived and designed the experiments, authored or reviewed drafts of the article, and approved the final draft.

The following information was supplied relating to ethical approvals (i.e., approving body and any reference numbers):

This study was conducted with the approval of Seoul National University Institutional Animal Care and Use Committee (SNUIACUC, approval number: SNU-181127-12).

The following information was supplied regarding data availability:

Raw data are available in the Supplemental Files.

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
