# Peer review of "Effect of sodium formate and lactic acid bacteria treated rye silage on methane yield and energy balance in Hanwoo steers"

_PeerJ, doi:10.7717/peerj.17920_

## Round 0.1 · original submission · Major Revisions

Dear Dr. Choi and colleagues:

Thanks for submitting your manuscript to PeerJ. I have now received three independent reviews of your work, and as you will see, the reviewers raised some concerns about the research. Despite this, these reviewers are optimistic about your work and the potential impact it will have on research studying ruminant biology and microbial interactions and impacts on ruminants. Thus, I encourage you to revise your manuscript, accordingly, considering all of the concerns raised by both reviewers.

Please revise your manuscript for clarity and limit jargon/reduce verbiage but be clear about the focus of the study and the target audience. Focus especially on clarity, and address the sections considered incomplete and/or unclear by the reviewers. It appears that certain key references are missing. The Methods should be clear, concise and repeatable. Please ensure this. Also, elaborate on the discussion of your findings, placing them within a broad and inclusive body of work by the field. Please supply any code or scripts in the supplemental material.

Please note that Reviewer 1 kindly provided a marked-up version of your manuscript.

I look forward to seeing your revision, and thanks again for submitting your work to PeerJ.

Good luck with your revision,

-joe

**Language Note:** PeerJ staff have identified that the English language needs to be improved. When you prepare your next revision, please either (i) have a colleague who is proficient in English and familiar with the subject matter review your manuscript, or (ii) contact a professional editing service to review your manuscript. PeerJ can provide language editing services - you can contact us at [email protected] for pricing (be sure to provide your manuscript number and title). – PeerJ Staff

·

Basic reporting

ID: peerj-reviewing-93247-v0-Manuscript
Title: Effect of sodium formate and lactic acid bacteria treated rye silage on ruminal fermentation
characteristics and methane yield of Hanwoo
This experiment was intriguing as it elucidated the profound impact of silage, elucidating the role of microorganisms and acids in facilitating optimal fermentation conditions. The insights gleaned, particularly concerning the utilization of energy in animals, are noteworthy. While the author adeptly addresses various aspects, there remain areas where the experimental findings could be critiqued constructively for the readers' benefit. I genuinely anticipate that upon implementing the suggested revisions, the authors will further enrich the scholarly discourse within this esteemed journal.

Abstract
• Line 35: "Lactobacillus plantarum" (italicized) and "1.5 x 10^10" (correctly formatted).

Results and Discussions
Regarding the discussion on experimental results, the authors should provide more focused critiques aligning with the principles established by the experimental data, ensuring clarity and relevance.
• In lines 228 to 233, inconsistencies with previously quoted information need correction. Also, reevaluation of the significance of dry matter (DM) differences as no statistical significance was found is necessary.
• Lines 236 to 238 require careful review to ensure consistency with the actual experimental results.
• Regarding lines 242 to 245, a precise explanation is needed regarding how the use of Na-FA leads to higher water-soluble carbohydrates (WSC) compared to other groups.
• In line 250, "ammonia nitrogen" should be standardized to "NH3-N" for consistency with abbreviations used throughout the manuscript.
• Regarding lines 259-263, the decrease in acetic acid after using the acidic additive requires a direct explanation.
• Lines 263-266 lack clarity in connecting the discussed event with the author's intended message. Further elucidation is necessary to convey the significance of this event within the context of the experiment.
• In line 273, "Lactobacillus species" should be italicized consistently with line 275.
• Lines 270-277 require clarity on the efficacy of using the Lactobacillus group, along with a precise definition of "nutrient loss" to avoid confusion.
• Line 280 needs reordering to maintain coherence, as it refers to both Table 2 and Table 3.
• Lines 281 to 282 cannot be interpreted correctly if explained this way. Control requires loss and Na-FA provision must have higher gross energy.
• Lines 284 to 285 should be reconciled with lines 275-277 to avoid contradictions.
• In line 288, "NH4-N" should be standardized to either "ammonia nitrogen" or "NH3-N" for consistency.
• Regarding line 289, specify the threshold for "High Level" and elaborate on the implications of ammonia nitrogen levels in the control group.
• Lines 296-297 need consistent usage of terms, either abbreviations or full words, for "lactic acid bacteria" and "sodium formate."
• Lines 300 to 301 should be revised for clarity and relevance beyond comparisons with the control group.
• Lines 329 to 339 fail to provide any elucidation on how Lactic Acid Bacteria and Sodium Formaldehyde impact Iso valeric Acid. Further elaboration is needed to address this gap in the discussion.
• In lines 334 to 335, discrepancies with actual results need correction to accurately reflect the data.
• Line 342 should abbreviate "lactic acid bacteria" and "sodium formate" for consistency.
• Regarding lines 360 to 363, avoid implying that mean data has better effects without statistical support.
• In line 376, abbreviate "lactic acid bacteria" and "sodium formate" for consistency.
• Lines 399 to 401 should be revised to reflect the lack of statistical difference more explicitly.

Experimental design

no comment

Validity of the findings

no comment

Additional comments

no comment

·

Basic reporting

After reviewing the paper "Effect of sodium formate and lactic acid bacteria-treated rye silage on ruminal fermentation characteristics and methane yield of Hanwoo," it is clear that this paper has the potential to make a significant contribution to the field of animal nutrition while also addressing environmental concerns. However, this paper is not publishable in its current state. However, I see the value in the research approach and recommend that the authors revise and resubmit their manuscript to the journal

Experimental design

• Material and method are unorganized, and I understand that the objective of the work that affects ensiling these additives on ensiling characterizes, rumen fermentation, and methane production of rye forage, and you mention in lines 83 and 84 that limited research regarding the effect of silage fermentation characteristics on the CH4 emissions of ruminants, even in recent studies, should be organized as follows:
1. Experiment 1: Ensiling Properties of Rye
1.1. Silage preparation
1.2. Fermentation profiles and chemical analysis
2. Experiment 2: Nutrtients digestibility, rumen fermentation characteries and methane production
2.1. Animals, experimental design and diet
2.2. Sampling and Data Collection
• The author should list the concentrate feed ingredients for each level of additive inclusion, as well as the chemical composition of the basal diet, and add them to Table 1. The sentence in line 152 should be moved to Section Silage Preparation.
• The author should mention the DMI measure and indicate measuring nutrient digestibility and, digestible energy and metaboilizable and net energy in material and method because it is mentioned in tables.
• Supplementary files should be listed in the section Material and Method.

Validity of the findings

• Since the discussion and results have been integrated, it is necessary to explain each result along with its interpretation. It is not feasible to explain all the results at once and then interpret them, as this may confuse the reader. For example, we can explain the results of PH level and then lactic acid production first, and then discuss mode of action or mechanism of these additives on lactic acid production and PH. Additionally, consider organizing them in a table format for clarity.

• The method of discussing the results includes many details that are unrelated to the topic; it is critical to explain the real reason for the outcome.
• To combine silage fermentation characteristics and chemical analysis, it is advisable to present them in a single table. When writing about materials and methods, arrange them in the order of the table. It is preferable to start with the fermentation characteristics of silage and then include the chemical analysis in a table and write and discuss them in a manuscript. Some studies suggest that the fermentation characteristics are related to the nutritional composition of the silage. For example, the pH value of the silage can impact fiber digestion during the ensiling process. An increase in crude protein in treated silage indicates a decrease in protein digestion rate, which is evident from reduction of the ammonia parameter. This approach will facilitate the interpretation of the results.
• As you discuss the fermentation parameters for silage, you must clarify whether each parameter in the study conforms to the specifications of good silage or not. For example, if the values of the PH of silage, acetic acid, butyric acid, propionic acid, and ammonia for the control and treatments are recommended for the good silage and indicative of its increase or decrease than the values recommended, you must discuss, explain, and discuss the mode of action of these additives on lower or higher parameter with references.
• In the second experiment conducted on animals, it is also necessary to record the results and provide their interpretation. It should be noted that if there are results that do not have an impact, such as in nutrient digestion, for example, it is essential to refer to references that either agree or disagree with the study and explain the reasons behind this discrepancy.
• It is critical to understand how silage fermentation quality and additive additions influence methane production in the rumen. This information is available in the references, which is especially important because it serves as the study's starting point and the foundation for the research. Furthermore, it explains how sodium formate reduces methane production in the rumen

Additional comments

Title
I recommend the authors change the title and include the study's main parameters as follows: "Ensiling characteristics, nutrient digestibility, fermentation characteristics, and methane production of additional sodium fumarate and lactic acid bacteria in rye forage.
Abstract:
- The abstract should be organized in the following order: study objective, animal number, experimental design, treatments, parameter measurement, and study findings.
- It does not find any results for silage fermentation characteristics; instead, it should explain them.

Introduction
- The work's goal is unclear; it should be clarified what the primary goals of using silage additives are, as well as how the novelties work. Thus, it suggested that the introduction should be organized as follows:
- A brief overview of the rye forage and its composition of the chemical and challenge of ensiling it as well as discuss the main challenges encountered during silage fermentation and their impact negatively on methane production in ruminants. A brief should show the negative impacts of methane production, both economic and environmental, on ruminants.
- The focus should be on microbial additives such as lactic acid bacteria and their role in improving silage quality, as well as their impact on rumen fermentation and methane production.
- Briefly discuss how acid additives improve silage quality, rumen fermentation, and methane production, as well as the issues that influence their use. So, sodium fumarate was used. By the way, the term acid-based additive should be changed to acid salt additives.
- it should write the hypothesis of work and objectives complete
• References must be updated
• Conclusions
Conclusions must be abbreviated while including important study findings and suggestions for the future.

Reviewer 3 ·

Basic reporting

This study investigated the effects of rye silage treated with sodium formate and lactic acid bacteria inoculant on the ruminal fermentation characteristics and methane yield of Hanwoo steers. The manuscript is based on a very good concept methodologically executed and well written especially resutls and discussion section. Some corrections needed to be addressed before its publication which are mentioned below.
1. Abstract section contained a large part of materials and methods, please reduce and talk more about the findings of this study.
2. Enteric CH4 production by livestock is big problem for climate crises, so i would suggest to write the problem first in introduction section followed by solution.
3. Line 228-230 and line 238 does not support your results as DM was not substantially different among treatments.
4. Table 1 and 2 contain chemical composition data twice, please represent only once.
5. Please show significance among treatments in Figure 3.

Experimental design

The experimental design is appropriate.

Validity of the findings

All data have been provided.

---

## Round 0.2 · Minor Revisions

Dear Dr. Choi and colleagues:

Thanks for revising your manuscript. The reviewers are very satisfied with your revision (as am I). Great! However, there remains a very minor issue to address per reviewer 2. Please address this ASAP so we may move towards acceptance of your work.

Best,

-joe

·

Basic reporting

After reviewing the paper " Effect of sodium formate and lactic acid bacteria 2 treated rye silage on ruminal fermentation 3 characteristics and methane yield of Hanwoo".The authors adequately addressed my feedback from the first round of peer review. I only have some minor comments for the final improvements

Experimental design

The Experimental design is clear and understandable

Validity of the findings

The author followed all existing comments, but the author should divide Experminat 1 and Experminat 2 in the result and discussion section, as shown in the material and method section.

Additional comments

Cancel the p value in section conclusion lines 518, 519, and 520.

Reviewer 3 ·

Basic reporting

Author has taken care of all the raised concerns, so I recommend publication of this manuscript.

Experimental design

Experimental design is appropriate.

Validity of the findings

Results could be interest of global community.

---

## Round 0.3 · accepted · Accept

Dear Dr. Choi and colleagues:

Thanks for once again resubmitting your manuscript to PeerJ. I now believe that your manuscript is suitable for publication. Congratulations! I look forward to seeing this work in print, and I anticipate it being an important resource for groups studying ruminant biology, as well as microbial interactions and their impacts on ruminants. Thanks again for choosing PeerJ to publish such important work.

Best,

-joe